# From Hospital to Home: Interdisciplinary Approaches to Optimise Palliative Care Discharge Processes

**DOI:** 10.3390/ijerph22071023

**Published:** 2025-06-27

**Authors:** Matthias Unseld, Timon Wnendt, Christian Sebesta, Jana van Oers, Jonathan Parizek, Lea Kum, Eva Katharina Masel, Pavol Mikula, Hans Jürgen Heppner, Elisabeth Lucia Zeilinger

**Affiliations:** 1Department of Clinical Research SBG, Academy for Ageing Research, Haus der Barmherzigkeit, 1160 Vienna, Austria; matthias.unseld@hb.at; 2Division of Palliative Medicine, Department of Medicine I, Medical University of Vienna, 1090 Vienna, Austria; christian.a.sebesta@meduniwien.ac.at (C.S.); lea.kum@meduniwien.ac.at (L.K.); eva.masel@meduniwien.ac.at (E.K.M.); 3Department of Clinical and Health Psychology, Faculty of Psychology, University of Vienna, 1010 Vienna, Austria; a12233653@unet.univie.ac.at; 4Medical Campus Upper Franconia (MCO), Friedrich-Alexander University Erlangen-Nuremberg, 91058 Erlangen, Germany; pavol.mikula@klinikum-bayreuth.de (P.M.); hans.heppner@fau.de (H.J.H.)

**Keywords:** discharge planning, end-of-life care, home-based care, hospital discharge, interview study, palliative care, qualitative research

## Abstract

The transition from hospital-based palliative care to home care is a critical phase often marked by logistical, medical, and emotional challenges. Effective discharge planning is essential to ensure continuity of care, yet gaps in communication, interdisciplinary coordination, and access to resources frequently hinder this process. This qualitative study explored key barriers, related support needs, and strategies for optimising palliative care discharge through semi-structured interviews with 28 participants, including healthcare professionals, recently discharged palliative care patients, and primary caregivers. Reflexive thematic analysis revealed five main themes: (1) discharge planning and coordination; (2) symptom management and medication; (3) psychosocial support; (4) communication and information; (5) the role of assistive devices and home care services. Discharge processes were often late or unstructured. Poor interdisciplinary collaboration and a lack of caregiver preparation also contributed to hospital readmissions and emotional distress. By focusing on needs, our analysis identifies not only what was lacking but also what is required to overcome these barriers. Our findings suggest that standardised discharge protocols and checklists, earlier planning, structured communication tools, and improved integration of home care services could enhance patient outcomes and reduce caregiver burden. Addressing psychosocial needs and ensuring timely access to assistive devices are also crucial. Strengthening interdisciplinary collaboration and refining discharge practices can facilitate smoother transitions and improve the quality of palliative care at home.

## 1. Introduction

Palliative care is an approach that focuses on improving the quality of life for patients facing life-threatening illnesses by addressing their complex physical, emotional, social, and spiritual needs [1]. As the global population continues to age and the prevalence of chronic diseases rises, the demand for palliative care services has grown significantly. According to estimates by the World Health Organization, approximately 4.4 million people in the WHO European Region require palliative care, a figure which is expected to increase as populations age [1]. This gap underscores the need to improve palliative care access, especially during the vulnerable transition from hospital to home. In Austria, demand for palliative care has steadily increased, yet national health reports continue to highlight regional disparities in access to home-based services and insufficient integration between hospital and community care. According to the Austrian Hospice and Palliative Forum, around 30% of patients in need of palliative care do not have timely access to coordinated home support, especially when moving between inpatient care settings [2].

The transition from inpatient palliative care to home care represents a vulnerable phase for patients and their families, often accompanied by significant logistical, medical, and emotional challenges. While many patients express a desire to spend their final phase of life at home, practical obstacles such as inadequate symptom management, lack of access to assistive devices, and insufficient home care resources can result in rapid hospital readmissions [3,4]. Studies suggest that up to 30% of readmissions for palliative care patients could be avoided with better discharge planning and community support [5]. Yet in practice, essential components of discharge management are often lacking. A study from Germany identified four quality criteria for discharge management—palliative care information transfer, discharge documentation, patient and caregiver counselling, and uninterrupted medication supply—but found that these were often missing from discharge records [6]. Addressing these deficiencies through effective discharge planning is essential to ensure a safe and supported transition to home care, ultimately reducing the burden on healthcare systems while improving patient outcomes.

Interdisciplinary collaboration is a cornerstone of palliative care. It involves a wide range of healthcare professionals, including doctors, nurses, social workers, dieticians and therapists. An effective discharge process requires their coordinated efforts to develop a holistic and patient-centred care plan [7].

However, research has consistently identified several barriers to achieving optimal discharge outcomes. Common challenges include a lack of standardised discharge protocols, insufficient communication between healthcare providers, delays in arranging home care services, and bureaucratic hurdles in accessing necessary equipment [3,8,9]. These challenges can leave patients and their families feeling overwhelmed and unprepared for the demands of home-based care.

In addition to logistical barriers, the transition to home care can impact psychological well-being [10]. Patients may experience heightened distress, fear, and apprehension due to the shift in their care environment [11]. Family members, who often serve as primary caregivers, can feel unprepared and burdened by the responsibilities associated with providing palliative care at home [12,13]. Despite these emotional needs, hospital discharge letters often focus on physical aspects while omitting information about social and psychological palliative domains [3]. This underscores the need for discharge planning that not only addresses clinical aspects, such as symptom management and medication, but also incorporates psychosocial support tailored to the emotional needs of both patients and caregivers. Despite widespread recognition of these issues, few studies have systematically explored the practical barriers and corresponding support needs for effective discharge from the perspectives of those most directly involved: patients, caregivers, and healthcare professionals. Most existing studies focus on singular viewpoints or rely on administrative data, overlooking the complex interplay of emotional, clinical, and logistical factors that shape the discharge experience. This qualitative interview study aims to fill this gap by identifying both the barriers encountered and the actionable needs required to overcome them in the discharge process, thereby offering actionable strategies for improving transitions to home-based palliative care. The findings will contribute to the development of standardised discharge protocols and improved coordination of care, with the aim of enabling patients to spend more time in their preferred environment and reducing the burden on carers and healthcare resources.

## 2. Methods

This study used a qualitative research design to obtain in-depth insights from multiple perspectives, including those of patients, family caregivers and healthcare professionals.

### 2.1. Study Population

The study involved three groups: (1) healthcare professionals (e.g., physicians, nurses, social workers, psychotherapists, physiotherapists and dieticians), with at least one year of experience in palliative care and direct involvement in the discharge process; (2) recently discharged patients (within 3 months); (3) their primary family caregivers. It was ensured that patients and caregivers were physically and cognitively able to give informed consent and participate in an interview.

### 2.2. Recruitment

The participants for the study were recruited from two sources: the palliative care department of the largest hospital in Austria, the General Hospital of Vienna, and mobile specialised palliative care teams operating in Vienna, but which are not part of the General Hospital. A purposive sampling strategy was employed to ensure that participants had direct, relevant experience with the palliative discharge process. Healthcare professionals were recruited through internal communications within the palliative care unit. Patients and caregivers were approached by the palliative care team during routine follow-up calls or home visits. All participants gave written informed consent prior to data collection. A total of 47 individuals were invited to participate: 20 healthcare professionals, 10 caregivers, and 17 patients. Of these, 28 agreed to participate—18 professionals, 5 caregivers, and 5 patients—yielding an overall participation rate of approximately 60%.

### 2.3. Procedure and Data Collection

Data were collected through semi-structured, in-depth interviews. An interview guide was developed based on published literature on the discharge process and the experience of the research team. The research team comprised palliative care clinicians, a qualitative research expert and a clinical psychologist. The interview guide included open-ended questions that allowed participants to give detailed, reflective responses and to focus on aspects that were relevant to them. Questions covered topics such as discharge planning, coordination of care, emotional impact, symptom management, and post-discharge support, challenges, and needs. The development process involved iterative team discussions among clinicians, psychologists, and qualitative researchers to ensure content validity and relevance across the different groups. The guide was piloted with two interviews, including a palliative care clinician and a palliative care nurse. Following the pilot interviews, the interview guide was refined.

Interviews were conducted either face-to-face in the hospital, in the patient’s home, or via video call, depending on the participant’s preference, health status, and logistical considerations. Interviewers were research team members with a medical background. All interviewers received interview and communication training prior to conducting the interviews. Interviews lasted between 30 and 50 min. All interviews were audio-recorded with participants’ consent and later transcribed verbatim. Field notes were also taken to capture non-verbal cues and contextual details.

### 2.4. Data Analysis

The data were analysed using reflexive thematic analysis [14]. After familiarisation with the data, initial codes were generated inductively, without pre-defined categories. The codes were grouped into themes and sub-themes. The coding tree and themes were subject to change throughout the coding process. Coding was conducted by a team of three researchers. All interviews were coded independently by two researchers. Discrepancies in coding were resolved through discussion within the coding team with the support of one of the lead researchers. The naming of themes and sub-themes was performed by the entire coding team. The analysis followed a reflexive approach, which views coding and theme development as active, interpretive processes. Coding was conducted collaboratively and iteratively, with team discussions used to deepen understanding, explore diverse perspectives, and ensure analytic rigour through reflexive dialogue. MAXQDA Analytics Pro software (2022) was used to facilitate coding, theme development and data management. Data collection continued until thematic saturation was reached, meaning no new themes or significant variations emerged from additional interviews. The research team collaboratively assessed saturation during coding and analysis discussions. We determined that further data collection was unlikely to yield novel insights relevant to our research aims.

In the case of patients and family caregivers, the inclusion of five individuals per group was sufficient to reach thematic saturation within these subgroups. This decision was also guided by ethical considerations: given the high physical and emotional burden on patients recently discharged from palliative care and their families, we aimed to minimise participant distress while still obtaining rich, relevant insights. Our sampling strategy, therefore, followed qualitative research principles of depth over breadth, using purposive sampling to ensure relevance and saturation.

### 2.5. Ethical Considerations

Participants were informed about the study’s objectives, and their right to withdraw at any point without consequences was clearly communicated. Written informed consent was obtained from all participants, ensuring their agreement to participate and have their interviews recorded. Anonymity and confidentiality were maintained by assigning pseudonyms to all participants, and all identifiable information was removed from transcripts. The study received ethical approval from the ethics committee of the Medical University of Vienna (No.1293/2022).

### 2.6. Researcher Reflexivity

All members of the research team brought disciplinary and professional experience, i.e., backgrounds in palliative medicine, clinical psychology, and qualitative research, which influenced how we engaged with and interpreted the data. Our shared familiarity with palliative care and a generally positive orientation toward its principles may have shaped how we approached and interpreted the data. All members of the research team viewed palliative care as a valuable and humane approach to end-of-life care. Additionally, our prior assumption that discharge processes are often problematic may have directed early attention toward challenges and systemic shortcomings. To mitigate potential bias, we engaged in regular team discussions to surface and critically examine these assumptions, allowing us to remain open to unexpected findings and alternative interpretations throughout the analysis.

## 3. Results

### 3.1. Sample

The final sample consisted of 28 people. This included seven doctors, five nurses, two social workers, one dietician, one physiotherapist, one occupational therapist, one psychotherapist, five patients and five primary family caregivers. Patients were aged between 60 and 82 years, including four men and one woman. Family carers were aged between 58 and 80 years, including two men and three women. Palliative care professionals were aged between 24 and 56 years. Their median experience in palliative care was 14.5 years. Participant characteristics are displayed in Table 1.

### 3.2. Results of Qualitative Analysis

The analysis generated five overarching themes, including various sub-themes, that capture meaningful patterns in how participants made sense of their experiences with the palliative care discharge process: (1) discharge planning and coordination; (2) symptom management and medication; (3) psychosocial support; (4) communication and information; (5) the role of assistive devices and home care services. An overview of the themes and subthemes is given in Figure 1.

#### 3.2.1. Theme 1: Discharge Planning and Coordination

The findings indicated that effective discharge planning is critical for a successful transition from hospital to home care. Participants reported several challenges and areas for improvement. The sub-theme *Timing and Standardisation* demonstrated a consensus that discharge planning often began too late in the patient’s hospital stay, which limited the time available to adequately address the needs of patients and families. One participant described their frustration with the abruptness of the discharge process: “*The worst thing was that… um…, during a ward round, the doctor suddenly announced that my mother would be discharged the next day. It was a shock.*” (P8). The lack of standardised discharge protocols was highlighted, with participants suggesting that involving the entire care team in discharge discussions could improve outcomes. This would include coordinating follow-up appointments, arranging home care services, and preparing the patient and family for the transition. The sub-theme *Role Clarity and Interdisciplinary Collaboration* emphasised the importance of a coordinated approach that integrates input from all relevant disciplines, such as medical professionals, social workers, and therapists, to create a comprehensive discharge plan. Effective role delineation within the team was identified as a necessary factor to ensure that all aspects of patient care were addressed. However, communication breakdowns were common, as one caregiver stated: “*The communication was completely lacking. Between the different teams. Between doctors, discharge management, and social services. Everyone was just focused on their own tasks.*” (P7).

#### 3.2.2. Theme 2: Symptom Management and Medication

Managing symptoms, particularly pain, emerged as a significant concern for patients transitioning to home care. Medication planning and ensuring continuity of symptom management were identified as key factors influencing patient outcomes. Within the sub-theme *Pain Management Challenges*, many patients experienced difficulties with pain control after discharge. The availability of clear instructions on medication regimens, including the use of opioids and other analgesics, was found to be inadequate. Participants suggested implementing individualised pain management plans and written action plans for patients and caregivers to follow, particularly for breakthrough pain. One nurse noted: “*Patients feel like they are being sent home with no plan. There’s uncertainty about how to get the necessary medications and whether prescriptions will be renewed on time*.” (P19). The sub-theme *Coordination with Outpatient Care* reflected gaps in the coordination of symptom management with outpatient providers. Participants recommended better communication and collaboration with home-based care teams to ensure seamless continuity of care. One patient expressed their struggle: “*Sometimes, it’s just a fundamental feeling of being overwhelmed when you come from a full-service environment in the hospital and are suddenly back home with all these things. How did they mean that again? Where do I need to order this now, or will it come automatically*?” (P3).

#### 3.2.3. Theme 3: Psychosocial Support

Psychosocial support was identified as an essential component of post-discharge care, addressing the emotional, social, and psychological challenges faced by patients and their families. In the sub-theme *Emotional Distress and Coping*, many patients reported experiencing anxiety, isolation, and emotional distress after discharge. It was also mentioned that family members felt overwhelmed by caring responsibilities. One physician described: “*A large part of the patients worry about being a burden on their relatives, and the fear of dependency on others is omnipresent. At the same time, there is concern about inadequate medical care*.” (P11). Participants highlighted the need for more structured psychosocial support services, including access to counselling, support groups, and regular mental health check-ins. The sub-theme *Reducing Caregiver Burden* included caregivers’ exhaustion and emotional distress. Successful discharge should include provisions for reducing their burden, such as access to counselling and respite care. Another sub-theme reflected the *Involvement of Social Workers and Psychologists*. This was seen as crucial in providing emotional support and practical assistance with navigating home care arrangements. Participants recommended that these professionals be integrated into the discharge planning process to help anticipate and address potential psychosocial challenges. A healthcare professional emphasised the following: “*Caregivers tell us they feel completely alone with the responsibility, especially at night when they don’t know who to call if something happens*.” (P26).

#### 3.2.4. Theme 4: Communication and Information

Effective communication emerged as a critical factor for successful discharge planning and home care. Within the sub-theme *Clear and Detailed Communication*, participants frequently reported that information about home care services, medication management, and follow-up care was insufficient or unclear. There was a preference for receiving both verbal explanations and written materials that could be referred to later. Ensuring that all essential information is conveyed in a structured and comprehensible manner was deemed necessary for empowering patients and families to manage care effectively. One healthcare professional described: “*Patients often feel left alone after discharge. They receive little guidance and sometimes only a brief explanation, which leads to insecurity*.” (P28). The sub-theme *Interdisciplinary and Patient-Centred Communication* found that inconsistencies in communication between different members of the healthcare team contributed to confusion and anxiety for patients and families. Patients and caregivers often reported feeling confused due to incomplete or rushed discharge meetings. Participants recommended adopting standardised communication tools, such as checklists or discharge summaries, to improve the consistency and clarity of information provided. A physician emphasised the need for improvements: “*We need to standardise the discharge process. Too often, patients and families receive conflicting information about what will happen next*.” (P15). *Involvement of Families and Caregivers*, another sub-theme, was also highlighted by participants. Families and carers need to be actively involved in discharge planning to ensure they understand their role in the patient’s home care.

#### 3.2.5. Theme 5: The Role of Assistive Devices and Home Care Services

The importance of assistive devices (e.g., hospital beds and wheelchairs) and professional home care services was underscored in supporting patients’ needs after discharge. In the sub-theme *Access and Availability*, participants identified delays in acquiring necessary assistive devices due to bureaucratic barriers, which often led to increased caregiver burden and patient discomfort. Essential equipment, such as hospital beds, mobility aids, and home nursing services, is critical for successful home care and should be provided as part of the discharge plan. The need for a more streamlined process to facilitate timely access to equipment was emphasised. As one nurse described: “*Many patients and relatives are overwhelmed with the task of organizing assistive devices. There needs to be more guidance from the hospital team to prevent rehospitalizations due to poor planning*.” (P13). Another sub-theme reflected on the *Home Care Team Integration*. This was seen as vital in managing the complexities of palliative care at home. Participants indicated that the availability of mobile palliative care teams was limited and recommended expanding these services to provide comprehensive support and reduce the burden on families. A physician noted: “*Although mobile palliative care teams receive predominantly positive feedback, there is still room for improvement in the availability of outpatient resources*.” (P17). Guidance from hospital staff in organising home care services and obtaining the necessary resources was also deemed essential.

## 4. Discussion

This study explored the challenges, needs, and strategies associated with the discharge process in palliative care, drawing on the perspectives of patients, family caregivers, and healthcare professionals. Our analysis identified and linked specific barriers in discharge practices with the corresponding needs required to overcome them. These included gaps in discharge planning and coordination, symptom management, psychosocial support, communication, and access to assistive devices and home care services. By presenting needs as responses to concrete obstacles, the findings offer both a diagnostic and solution-oriented view of the discharge process. Addressing these gaps is essential to ensure a safe and supported transition from hospital to home care. The findings highlight the need for early, structured discharge planning. Without standardised protocols and interdisciplinary coordination, transitions were often rushed and poorly managed. This aligns with prior research, emphasising the importance of clear roles and teamwork in palliative care discharge [15]. Standardised checklists and early multidisciplinary planning could improve preparedness and reduce hospital readmissions [16,17]. In addition to a standardised discharge checklist, providing structured discharge summaries and written materials alongside verbal explanations could improve the clarity and quality of post-discharge care [18]. Clear communication should remain a key focus, and ongoing studies are currently exploring this topic in greater depth [19].

A major concern raised by participants was the challenge of maintaining effective symptom control post-discharge, particularly in pain management. This reflects existing research highlighting the difficulties of ensuring medication continuity and adequate pain relief at home (e.g., Gnass et al., 2018 [20]; Isenberg et al., 2021 [11]). Other studies have also highlighted the challenges of managing non-pain and non-physical symptoms in home care [16,21]. Non-physical symptoms such as loss of dignity have been linked to an increased desire to hasten death, which underscores the need to prioritise their management in palliative care [22]. Individualised palliative care and symptom management plans—with clear written instruction—can help patients and caregivers feel more prepared [23]. Our data indicates a lack of collaboration and communication between hospital teams and mobile palliative care providers. Strengthening coordination between these groups is critical to ensure seamless medication management, delivering high-quality care [17,24,25], and adequate access to palliative care at home, a priority in palliative care practice and research [26].

Access to assistive devices and home care services was another significant barrier to effective palliative care at home. Delays in obtaining essential equipment, such as hospital beds and mobility aids, contributed to increased caregiver burden and patient discomfort. Streamlining the process of securing necessary resources prior to discharge is essential to ensure a smooth transition [25]. Improving the availability and coordination of mobile palliative care teams could further support home care and prevent unnecessary hospital readmissions.

The emotional burden of transitioning to home-based palliative care was evident in both patient and caregiver accounts. Many patients expressed fears of being a burden, while caregivers often felt unprepared and overwhelmed. These findings align with prior research indicating that caregivers of palliative patients experience high levels of distress and burnout [27]. Expanding access to psychosocial support services, such as counselling, respite care, and structured caregiver training, could mitigate these challenges [13]. A randomised controlled trial found that an online intervention significantly improved caregivers’ quality of life, despite no change in caregiver burden. This suggests that caregiver support is essential for well-being, regardless of the actual burden of caregiving [27]. Involving social workers and psychologists in the discharge planning process could provide essential emotional and practical support for families. A key concern should be to reduce patient loneliness and ensure psychological safety in the final stages of life.

While many of the identified issues have been reported in previous literature, our study provides new contributions in several key areas. First, by including the voices of patients, caregivers, and diverse healthcare professionals, we present a comprehensive view of how discharge planning functions (and often fails) in practice. Second, our findings offer concrete, real-world examples of how they unfold, such as caregivers being informed about discharge decisions with little notice or professionals facing unclear role responsibilities. Third, the study not only confirms known gaps but also links them directly to participant-informed strategies that are both practical and context-specific, providing a basis for implementing meaningful change in discharge practices.

### 4.1. Recommendations

Based on these findings, several recommendations emerge (see also Figure 2). Discharge planning should start earlier and follow standardised protocols to ensure comprehensive coordination across all disciplines, including medication continuity, symptom management, and caregiver education. Individualised pain management plans and seamless collaboration with outpatient and home-based care teams are essential for effective symptom control. Psychosocial support should be strengthened through counselling, support groups, and respite care options, with increased involvement of social workers and psychologists. Communication strategies must be improved by using standardised discharge summaries, checklists, and actively involving families and caregivers in planning. To facilitate home-based care, processes for accessing assistive devices should be streamlined, and mobile palliative care teams should be expanded to enhance support after discharge.

### 4.2. Limitations

This study offers valuable insights into the palliative care discharge process, yet several limitations should be acknowledged. First, the sample size was geographically limited to Vienna, Austria. This constrains the generalisability of the findings to other healthcare systems with different organisational structures or levels of community-based palliative care provision. Second, while efforts were made to include multiple perspectives, the number of patient and caregiver participants was modest. This may result in an overrepresentation of professional viewpoints, particularly in shaping the identified themes. Third, the use of qualitative interviews, while rich in contextual detail, relies on participants’ retrospective accounts. This introduces the possibility of recall bias or selective memory, especially given the emotional complexity of end-of-life care. Moreover, participants who chose not to take part may hold systematically different views, introducing potential selection bias. Fourth, although thematic analysis was conducted by a multidisciplinary team with reflexive practices in place, the interpretative nature of qualitative coding always carries some degree of subjectivity. While steps were taken to ensure inter-coder reliability and analytical rigour, complete objectivity cannot be assumed. Finally, the scope of the results is largely descriptive and context-specific. While actionable recommendations are offered, the study was not designed to test the effectiveness of specific interventions. Further research, particularly longitudinal and interventional studies, is needed to evaluate the impact of proposed strategies on patient and caregiver outcomes.

## 5. Conclusions

The results of this study highlight the urgent need for improved discharge planning, interdisciplinary coordination, and post-discharge support in palliative care. By addressing the gaps in symptom management, psychosocial support, communication, and access to home care resources, healthcare providers can improve patients’ quality of life and reduce caregiver burden. Implementing standardised protocols and expanding support services are essential steps to optimise palliative care transitions and ensure that patients can spend the last phase of their lives in their preferred environment with dignity and comfort.

## Figures and Tables

**Figure 1 ijerph-22-01023-f001:**
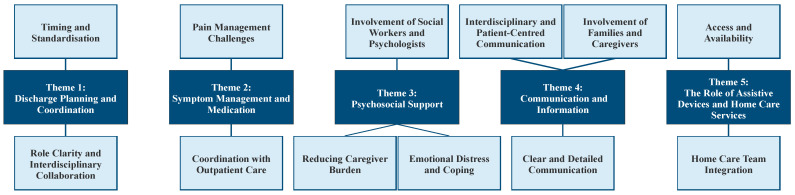
Themes and sub-themes derived from interviews.

**Figure 2 ijerph-22-01023-f002:**
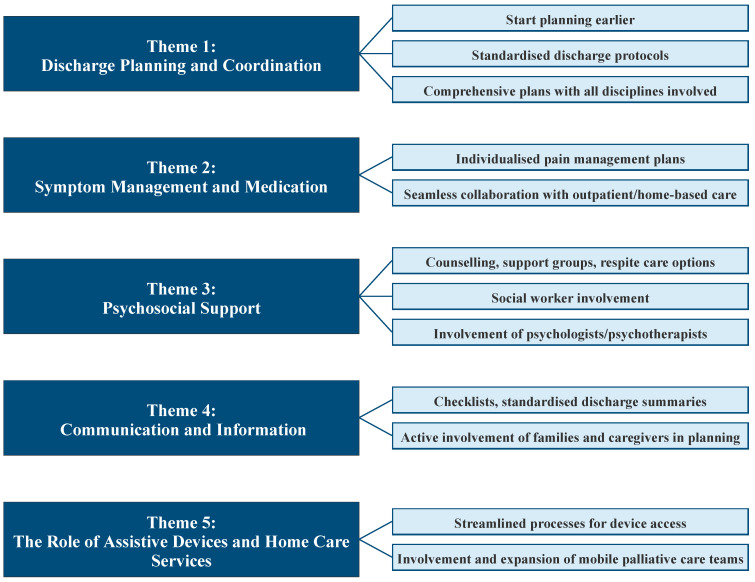
Recommendations derived from interviews.

**Table 1 ijerph-22-01023-t001:** Sample characteristics.

Participant Number	Age	Role
P1	72	patient
P2	65	patient
P3	60	patient
P4	78	patient
P5	82	patient
P6	58	relative
P7	61	relative
P8	68	relative
P9	74	relative
P10	80	relative
P11	24	physician
P12	28	nurse
P13	32	nurse
P14	36	physician
P15	39	physician
P16	41	nurse
P17	44	physician
P18	48	physician
P19	50	nurse
P20	53	physician
P21	55	nurse
P22	56	physician
P23	30	Interdisciplinary professionals
P24	35	Interdisciplinary professionals
P25	40	Interdisciplinary professionals
P26	45	Interdisciplinary professionals
P27	50	Interdisciplinary professionals
P28	55	Interdisciplinary professionals

## Data Availability

The raw data supporting the conclusions of this article will be made available by the authors upon request.

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
