# Peer review of "From Hospital to Home: Interdisciplinary Approaches to Optimise Palliative Care Discharge Processes"

_ijerph, 2025, doi:10.3390/ijerph22071023_

Round 1
Reviewer 1 Report
Comments and Suggestions for Authors
This is a well-written manuscript addressing an important clinical issue - the challenges you identify around discharge planning are pressing and improvement is clearly warranted. Your introduction establishes that discharge planning should incorporate both clinical and psychosocial elements (lines 78-80). Your findings appear to confirm this existing knowledge (lines 269-271) rather than generate new insights. What specific aspects of these processes were previously unknown and warranted investigation? Without understanding what's keeping these problems in place, it's perhaps difficult to see how confirming what is already known will lead to real change in practice.
There appears to be a mismatch between your stated methodology and analytical approach. Braun and Clarke emphasise that thematic analysis is fundamentally interpretive and meaning-focused, yet your analysis prioritises inter-coder reliability and treats participant subjectivity as potential bias rather than valuable interpretive data (line 338-9). The 'different perspectives' (line 83) appear to function primarily as sampling categories rather than representing genuine exploration of how different stakeholders make meaning of discharge experiences. Additionally, the manuscript lacks any reflexive engagement with the researchers' interpretive role or acknowledgement of how your perspectives shaped theme development - a crucial component of rigorous thematic analysis.
Your results present categorical findings rather than developed themes that capture meaningful patterns in how participants make sense of their experiences. You might consider whether content analysis or framework analysis would better align with your actual analytical approach and research aims.
Author Response
REVIEWER 1:
This is a well-written manuscript addressing an important clinical issue - the challenges you identify around discharge planning are pressing and improvement is clearly warranted. Your introduction establishes that discharge planning should incorporate both clinical and psychosocial elements (lines 78-80). Your findings appear to confirm this existing knowledge (lines 269-271) rather than generate new insights. What specific aspects of these processes were previously unknown and warranted investigation? Without understanding what's keeping these problems in place, it's perhaps difficult to see how confirming what is already known will lead to real change in practice.
We thank the reviewer for this thoughtful and constructive feedback. While it is true that previous literature has highlighted the importance of clinical and psychosocial elements in discharge planning, our study contributes novel insights in three specific ways:
- Triangulated perspectives: Few studies to date have systematically gathered and compared the experiences of patients, family caregivers, and interdisciplinary healthcare professionals involved in palliative discharge. This triangulation allowed us to identify not only that gaps exist, but where and why they persist across system levels—from misaligned communication within teams to unanticipated emotional burdens on families.
- Understanding how problems unfold in practice: While the challenges we identified (such as delayed planning or communication gaps) are not entirely new, our study provides concrete, detailed examples of how these issues are experienced by different stakeholders in real settings. For instance, caregivers reported being informed of discharge decisions without warning, and professionals described role confusion during coordination. These real-world accounts help reveal why these problems persist, despite widespread recognition.
- Translation to practice: By linking each barrier to a clearly expressed need or strategy (e.g., integrating social workers earlier, using written action plans for medication), we provide participant-informed suggestions for intervention. In this way, our study does not merely confirm previous findings, but helps bridge the gap between acknowledging challenges and implementing improvements.
We have now clarified this contribution more explicitly in the Discussion section by adding a paragraph right before the Recommendations Section. We hope this explanation addresses the reviewer’s concern and clarifies the added value of our findings.
There appears to be a mismatch between your stated methodology and analytical approach. Braun and Clarke emphasise that thematic analysis is fundamentally interpretive and meaning-focused, yet your analysis prioritises inter-coder reliability and treats participant subjectivity as potential bias rather than valuable interpretive data (line 338-9). The 'different perspectives' (line 83) appear to function primarily as sampling categories rather than representing genuine exploration of how different stakeholders make meaning of discharge experiences. Additionally, the manuscript lacks any reflexive engagement with the researchers' interpretive role or acknowledgement of how your perspectives shaped theme development - a crucial component of rigorous thematic analysis.
Your results present categorical findings rather than developed themes that capture meaningful patterns in how participants make sense of their experiences. You might consider whether content analysis or framework analysis would better align with your actual analytical approach and research aims.
Thank you for this thoughtful and insightful comment. We acknowledge the mismatch highlighted between the theoretical underpinnings of reflexive thematic analysis as outlined by Braun and Clarke and the way our analytical process was initially described. We agree that thematic analysis is inherently interpretive, and our original description inadequately conveyed our reflexive engagement with the data and our own role as researchers in theme development.
To address your concern, we have revised the Data Analysis section to clarify that while multiple coders were involved, the aim was not to achieve inter-coder reliability in a positivist sense. Rather, our discussions were part of a reflexive process that allowed us to explore different interpretations and co-construct meaning through dialogue. Furthermore, we added a new section on Researcher Reflexivity that reflects on how our professional backgrounds and disciplinary lenses influenced the analysis, in line with Braun and Clarke’s emphasis on transparency and positionality.
We respectfully maintain that thematic analysis remains the most appropriate analytical approach for our research aims, given our interest in capturing shared meanings and interpretive depth across stakeholder experiences.
Reviewer 2 Report
Comments and Suggestions for Authors
Introduction
The text contains long and sometimes repetitive sentences, which may affect clarity. It is recommended to break down lengthy sentences and avoid frequent use of passive constructions and successive enumerations without pauses. For example, the sentence “Interdisciplinary collaboration is a cornerstone of palliative care...” could be split to improve readability and emphasize key points.
Although the study’s rationale is well outlined, it is advisable to expand on the description of the knowledge gap, making it clearer why existing studies are insufficient and how this qualitative study can help address this gap. This would strengthen the research question.
While the text includes relevant international references, it would be beneficial to incorporate, if possible, data or studies from the local context (i.e., the country or region where the study was conducted), which would enhance contextualization and the applicability of the findings.
Methodology
Clarity and conciseness in population description (2.1)
Some sentences are fragmented and contain unnecessary repetition.
Example: “First, healthcare professionals working in palliative care. This included...”
Suggestion: Combine into one sentence:
“The study included three groups: (1) healthcare professionals (e.g., physicians, nurses, social workers, psychotherapists, physiotherapists, and dieticians) with at least one year of experience in palliative care and direct involvement in the discharge process; (2) palliative care patients recently discharged (within 3 months); and (3) their primary family caregivers.”
Recruitment (2.2)
The description is functional but could provide more detail regarding the number of people invited and the participation rate.
Suggestion: Add the following:
“A total of X participants were approached, and Y agreed to participate, yielding a participation rate of Z%.”
Procedure and data collection (2.3)
The identity of the interviewers is briefly mentioned, but their background may influence the responses.
Suggestion: Specify interviewers' training or prior experience with qualitative interviews.
Example:
“Interviewers had prior experience conducting qualitative interviews and received specific training to ensure consistency and sensitivity during data collection.”
Data analysis (2.4)
The description of thematic analysis is appropriate, but it lacks a reference to theoretical saturation, which is essential in qualitative studies.
Suggestion: Include whether saturation was reached and how it was assessed.
Example:
“Data collection continued until thematic saturation was reached, meaning no new relevant themes emerged in subsequent interviews.”
Author Response
Introduction
The text contains long and sometimes repetitive sentences, which may affect clarity. It is recommended to break down lengthy sentences and avoid frequent use of passive constructions and successive enumerations without pauses. For example, the sentence “Interdisciplinary collaboration is a cornerstone of palliative care...” could be split to improve readability and emphasize key points.
Thank you for this comment. We have revised the indicated sentence and made further changes to lengthy sentences and passive constructions to enhance readability throughout the manuscript.
Although the study’s rationale is well outlined, it is advisable to expand on the description of the knowledge gap, making it clearer why existing studies are insufficient and how this qualitative study can help address this gap. This would strengthen the research question.
We expanded on the existing gap and emphasised the added value of the present study more clearly at the end of the Introduction section. Furthermore, we added a paragraph to the discussion section before the recommendations section to emphasise the importance of the present research.
While the text includes relevant international references, it would be beneficial to incorporate, if possible, data or studies from the local context (i.e., the country or region where the study was conducted), which would enhance contextualization and the applicability of the findings.
We agree with the reviewer and have added the following paragraph which reflects on the situation in Austria: “In Austria, demand for palliative care has steadily increased, yet national health reports continue to highlight regional disparities in access to home-based services and insufficient integration between hospital and community care. According to the Austrian Hospice and Palliative Forum, around 30% of patients in need of palliative care do not have timely access to coordinated home support, especially when moving between inpatient care settings (GOEG, 2023).”
Methodology
Clarity and conciseness in population description (2.1)
Some sentences are fragmented and contain unnecessary repetition.
Example: “First, healthcare professionals working in palliative care. This included...”
Suggestion: Combine into one sentence:
“The study included three groups: (1) healthcare professionals (e.g., physicians, nurses, social workers, psychotherapists, physiotherapists, and dieticians) with at least one year of experience in palliative care and direct involvement in the discharge process; (2) palliative care patients recently discharged (within 3 months); and (3) their primary family caregivers.”
Thank you for this suggestion. We amended the description accordingly.
Recruitment (2.2)
The description is functional but could provide more detail regarding the number of people invited and the participation rate.
Suggestion: Add the following:
“A total of X participants were approached, and Y agreed to participate, yielding a participation rate of Z%.”
We added the following to the recruitment section: “A total of 47 individuals were invited to participate: 20 healthcare professionals, 10 caregivers, and 17 patients. Of these, 28 agreed to participate—18 professionals, 5 caregivers, and 5 patients—yielding an overall participation rate of approximately 60%.”
Procedure and data collection (2.3)
The identity of the interviewers is briefly mentioned, but their background may influence the responses.
Suggestion: Specify interviewers' training or prior experience with qualitative interviews.
Example:
“Interviewers had prior experience conducting qualitative interviews and received specific training to ensure consistency and sensitivity during data collection.”
We included the following in the procedure and data collection section: “All interviewers received interview and communication training prior to conducting the interviews.”
Furthermore, we added a paragraph on researcher reflexivity, which provides a more in-depth description of the research team's background.
Data analysis (2.4)
The description of thematic analysis is appropriate, but it lacks a reference to theoretical saturation, which is essential in qualitative studies.
Suggestion: Include whether saturation was reached and how it was assessed.
Example:
“Data collection continued until thematic saturation was reached, meaning no new relevant themes emerged in subsequent interviews.”
At the end of the data analysis section, we included the following: “Data collection continued until thematic saturation was reached, meaning no new themes or significant variations emerged from additional interviews. The research team collaboratively assessed saturation during coding and analysis discussions. We determined that further data collection was unlikely to yield novel insights relevant to our research aims.”
Reviewer 3 Report
Comments and Suggestions for Authors
Dear authors,
Reading and reviewing your work was not just an excellent opportunity to learn, but also a privilege to delve into the significant contributions you are making to the field of palliative care.
Your research has successfully addressed an essential aspect in the life trajectory of patients with palliative care needs. This is a significant achievement, and I congratulate you on it!
I describe my considerations below:
1) Abstract: The abstract is well-written, but it could be more concise and provide a clearer overview of the research.
2) Introduction:
2.1) Well written. It reveals the research problem and the gap you intend to fill. However, I believe that instead of using the WHO's 2002 definition of palliative care, they should use its 2023 adaptation;
2.2) In the final paragraph, there is a significant inconsistency. First, you state that the study aims to explore the barriers to effective discharge from the patient's perspective. Later, you mention that you intend to identify the challenges and unmet needs in the transition process. These are two different objectives, and they imply different study methodologies; 3) Methodology
3.1) The inclusion criteria state that all subjects should be physically and cognitively able to participate. These criteria are unclear and too broad. These criteria should not apply to health professionals, who must be physically and cognitively fit to work.
3.2) The recruitment sources are described as being different. However, from what we can see, they appear to belong to the same institution or service. So why separate them?
3.2) The size of the sample required and how to achieve it are not indicated.
3.3) The use of a script for the interview is mentioned, but nothing is said about its composition and how it was developed. Therefore, it does not allow for replicability and comparability.
3.4) It would be helpful to explain why only 10 patients/family members were included in the description of the participants.
3.5) In Table 1, the role of participants P23-P27 is indicated as interdisciplinary professionals. It should be clear what they are specifically.
3.6) They indicate 14.5 years as a measure of central tendency for the length of experience in palliative care. How did they choose the mean? You did not analyze normality in such a small sample. The most appropriate approach here is to present the median.
3.7) In the analysis of qualitative data, they only address the needs. And what about the barriers that were in the initial objective?
4) Discussion
4.1) The focus of the discussion shifts back to the needs, whereas at the beginning of the work, the barriers were the primary objective. They will have to clearly define what is desired and then focus on the choice.
5) Limitations
5.1) The limitations section is quite generic. It would be beneficial to address specific methodological weaknesses and the breadth of the results to provide a more comprehensive review of the research.
Congratulations.
Thank you for your hard work and dedication to this research.
Keep up the good work.
Author Response
Dear authors,
Reading and reviewing your work was not just an excellent opportunity to learn, but also a privilege to delve into the significant contributions you are making to the field of palliative care.
Your research has successfully addressed an essential aspect in the life trajectory of patients with palliative care needs. This is a significant achievement, and I congratulate you on it!
Thank you for this positive evaluation of our work. We really appreciate your encouragement.
I describe my considerations below:
1) Abstract: The abstract is well-written, but it could be more concise and provide a clearer overview of the research.
We have revised the abstract to make it more concise and to provide a clearer summary of the study’s aims, methods, key findings, and implications.
2) Introduction:
2.1) Well written. It reveals the research problem and the gap you intend to fill. However, I believe that instead of using the WHO's 2002 definition of palliative care, they should use its 2023 adaptation;
Thank you for this helpful remark. We changed the reference accordingly.
2.2) In the final paragraph, there is a significant inconsistency. First, you state that the study aims to explore the barriers to effective discharge from the patient's perspective. Later, you mention that you intend to identify the challenges and unmet needs in the transition process. These are two different objectives, and they imply different study methodologies;
We agree with the reviewer and have revised the last paragraph to more accurately reflect the objectives of our study.
3) Methodology
3.1) The inclusion criteria state that all subjects should be physically and cognitively able to participate. These criteria are unclear and too broad. These criteria should not apply to health professionals, who must be physically and cognitively fit to work.
Thank you for this comment. We revised the entire paragraph on Study population and inclusion criteria for more clarity.
3.2) The recruitment sources are described as being different. However, from what we can see, they appear to belong to the same institution or service. So why separate them?
We clarified the description, which now reads: “ The participants for the study were recruited from two sources: the palliative care department of the largest hospital in Austria, the General Hospital of Vienna, and mobile specialised palliative care teams operating in Vienna, but which are not part of the General Hospital.”
3.2) The size of the sample required and how to achieve it are not indicated.
As to the sample size, we added a description of our approach to saturation in the Data Analysis section.
3.3) The use of a script for the interview is mentioned, but nothing is said about its composition and how it was developed. Therefore, it does not allow for replicability and comparability.
We have revised this description to provide more detail about the development and content of the interview guide.
3.4) It would be helpful to explain why only 10 patients/family members were included in the description of the participants.
Only 5 patients and 5 family caregivers participated in the study.
3.5) In Table 1, the role of participants P23-P27 is indicated as interdisciplinary professionals. It should be clear what they are specifically.
Thank you for your feedback. We have decided not to disclose this information. The reason for this is that the location of the study is known, as is the fact that it was conducted on the palliative care ward. Disclosure of these professions, of which only a few work at the clinic (e.g. dieticians and psychotherapists), would be a risk as it would allow the identification of individuals and would not ensure anonymity. In chapter “3.1. Sample”, we list the profession of the persons belonging to the interdisciplinary professionals.
3.6) They indicate 14.5 years as a measure of central tendency for the length of experience in palliative care. How did they choose the mean? You did not analyze normality in such a small sample. The most appropriate approach here is to present the median.
We computed the median, which was 14.5 years (a value given by one participant). We added this information in the Sample section. It now reads: “Their average experience in palliative care, as determined by the median, was 14.5 years.”
3.7) In the analysis of qualitative data, they only address the needs. And what about the barriers that were in the initial objective?
We thank the reviewer for this insightful observation. We agree that identifying barriers was a key part of our original research aim. In our thematic analysis, barriers were indeed captured and described. However, these barriers are closely interwoven with the needs and strategies identified by participants, and we chose to focus on what is required to overcome these obstacles to generate actionable insights for practice improvement. For example, the need for earlier discharge planning arises from the barrier of delayed coordination; the call for better psychosocial support reveals the emotional burden and isolation caregivers described.
Our intention was to go beyond identifying challenges by highlighting what can be done to overcome them. We believe that this approach not only addresses the barriers but also provides a more meaningful and constructive contribution to improving practice.
Nonetheless, we appreciate that this distinction may not have been sufficiently clear in the manuscript. We have therefore made the following revisions to clarify our approach:
- In the Abstract and the end of the Introduction, we now more explicitly state that our themes include both barriers and needs, that they are interwoven, and that our focus was set on the needs and strategies to improve the current situation.
- In the Discussion, we now open with a summary noting that both barriers and corresponding needs were identified and linked.
We hope these clarifications address the reviewer’s concern.
4) Discussion
4.1) The focus of the discussion shifts back to the needs, whereas at the beginning of the work, the barriers were the primary objective. They will have to clearly define what is desired and then focus on the choice.
Thank you again for this comment. We realise that the manuscript was unclear about the barriers, needs and actionable strategies. We have addressed these issues, as outlined in our response to your comment 3.7 above.
5) Limitations
5.1) The limitations section is quite generic. It would be beneficial to address specific methodological weaknesses and the breadth of the results to provide a more comprehensive review of the research.
We agree with the reviewer and have revised the entire limitations section to include more specific methodological constraints, including sample composition, potential biases introduced by the qualitative design, and the limited generalisability of our findings beyond the local healthcare context. We have also clarified the scope and descriptive nature of our results.
Congratulations.
Thank you for your hard work and dedication to this research.
Keep up the good work.
Thank you for taking the time to provide feedback on our work.
Round 2
Reviewer 3 Report
Comments and Suggestions for Authors
Dear authors
I want to begin by acknowledging your significant effort, openness, and flexibility in considering the points I raised during the initial review. Your dedication to this research is commendable.
Your positive response to the initial review is encouraging, and I appreciate your efforts in considering all the points raised.
While many of the points were addressed, there are still two considerations that require your attention.
They are:
3.4) It would be helpful to explain why only 10 patients/family members were included in the description of the participants.
Your answer: “Only five patients and five family caregivers participated in the study.”
My comment: your answer does not answer the request. You do not explain why only 10 patients/family members were included in the study. You answer without explaining.
3.6) You indicate 14.5 years as a measure of central tendency for the length of experience in palliative care. How did they choose the mean? You did not analyse normality in such a small sample. The most appropriate approach here is to present the median. Your response: “We computed the median, which was 14.5 years (a value given by one participant). We added this information in the Sample section. It now reads: “Their average experience in palliative care, as determined by the median, was 14.5 years.”
My comment: If you present the median correctly, you should not say that the median determines the average experience. The median and average are two different measures. You should correct it.
Good effort.
Best regards
Author Response
Dear authors
I want to begin by acknowledging your significant effort, openness, and flexibility in considering the points I raised during the initial review. Your dedication to this research is commendable.
Your positive response to the initial review is encouraging, and I appreciate your efforts in considering all the points raised.
Thank you for your positive and encouraging comments, and for taking the time to review our manuscript.
While many of the points were addressed, there are still two considerations that require your attention.
They are:
3.4) It would be helpful to explain why only 10 patients/family members were included in the description of the participants.
Your answer: “Only five patients and five family caregivers participated in the study.”
My comment: your answer does not answer the request. You do not explain why only 10 patients/family members were included in the study. You answer without explaining.
Thank you for your comment. We acknowledge that our previous response lacked sufficient explanation. We now have included the following in the Methods Section: “In the case of patients and family caregivers, the inclusion of five individuals per group was sufficient to reach thematic saturation within these subgroups. This decision was also guided by ethical considerations: given the high physical and emotional burden on patients recently discharged from palliative care and their families, we aimed to minimize participant distress while still obtaining rich, relevant insights. Our sampling strategy therefore followed qualitative research principles of depth over breadth, using purposive sampling to ensure relevance and saturation.”
3.6) You indicate 14.5 years as a measure of central tendency for the length of experience in palliative care. How did they choose the mean? You did not analyse normality in such a small sample. The most appropriate approach here is to present the median. Your response: “We computed the median, which was 14.5 years (a value given by one participant). We added this information in the Sample section. It now reads: “Their average experience in palliative care, as determined by the median, was 14.5 years.”
My comment: If you present the median correctly, you should not say that the median determines the average experience. The median and average are two different measures. You should correct it.
We apologize for not addressing your comment properly. We have corrected the wording of the sentence, which now reads: “Their median experience in palliative care was 14.5 years.”
Good effort.
Best regards